# OpenGLA: Topology and Task Adaptive ~~Foundation~~ Model for Power System Graph-Language Answering

## Abstract

Foundation models have shown impressive cross-modal generation and problem-solving abilities, yet directly applying them to power systems remains challenging due to strict requirements on accuracy, efficiency, and physical interpretability. We propose OpenGLA, a distinct graph–language foundation model for power systems. OpenGLA encodes grid states with a topology-adaptive GCN and an adaptive nodal feature encoder, projects them into the language embedding space, and fuses with textual instructions via a Mixture-of-Transformers module. A lightweight transformer detokenizer is designed to enable precise floating-point outputs. Experiments demonstrate that OpenGLA achieves superior multi-task and cross-topology performance with one-time training on different task-topology pairs, establishing a promising architecture for future large-scale power system foundation models.

## 1 Introduction

Power-system operations span diverse tasks—forecasting, state estimation, anomaly detection, OPF—fed by highly heterogeneous data (PMU/SCADA time series, topology and asset metadata, renewable and load forecasts, market bids, operation logs). Although recent task-specific AI solutions have advanced these problems, case-by-case development requires strenuous, domain-expert efforts and can not readily transfer across grids or tasks. Breakthroughs in general-purpose foundation models—e.g., GPT (Brown et al., 2020; Achiam et al., 2023), Gemini (Team et al., 2023; 2024; Comanici et al., 2025), and DeepSeek (Liu et al., 2024a; Guo et al., 2025)—have demonstrated impressive generalized cross-modality question-answering and problem-solving performance. These models stimulate research on domain-specific foundation models, including medicine (Lee et al., 2020), law (Zhou et al., 2024), and finance (Wu et al., 2023). It is worth exploring whether these models can trigger industrial improvements and bring significant economic and social benefits, which is particularly noteworthy in the electric energy sector.

Several recent efforts focused on the development of foundation models for the domain of electric power systems. Different from the general question-answering scenarios, task solving in specific fields like power systems brings strict requirements of accuracy and physical understanding. Directly applying general foundation models typically leads to unsatisfying response quality (Majumder et al., 2024), or impractical toy examples in key operational tasks (Huang et al., 2024). Some other research has leveraged the powerful coding capabilities of the foundational model to build agents that generate executable code to operate specialized power system software through MCP (Zhang & Xie, 2025). However, these approaches do not alleviate the speed bottleneck inherent in the specialized software itself, which is exactly the previous task-specific AI methods were built for. In addition, some

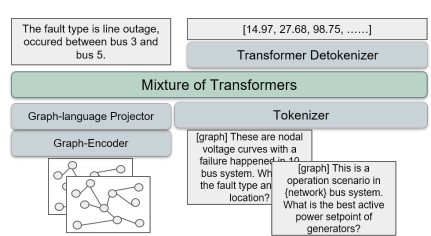

Figure 1: **OpenGLA workflow example on fault detection and optimal power flow.** [graph] represents the placeholder for graph embeddings in the language model processing. The language instruction indicates what task and physical system we are dealing with. The answer is formatted in natural language or detokenized as floating-point vectors upon tasks.

Figure 2: **Graph-Language Power Foundation Model Architecture**. **(A) Graph-language cross-modality model.** The graph data $X_g$ and language prompts $X_l$ are processed by the graph encoder and language tokenizer to get graph embedding $H_g$ and language embedding $H_l$ correspondingly, where the graph embedding $H_g$ is projected to the language hidden space via the projector $W$. The embeddings are fed into an Mixture-of-Transformers (MoT) model to get fused embedding for the downstream language or floating response $X_a$. The floating responses are generated using a light-weight transformer detokenizer. **(B) Nodal feature encoder.** We design a two-layer 1D-convolution layer with GeLU activation and residual connections to encode the task-varied nodal features. An adaptive average pooling layer is utilized to output fixed-shape embeddings. **(C) Graph Encoder.** We utilize the topology adaptivity of GCN to achieve adaptivity to different underlying power grids. We use a 3-layer GCN with GeLU activation and residual connections to encode the concatenated nodal embeddings as graph embeddings. **(D) Mixture-of-Transformers (MoT) details**. The graph and language modalities use separate transformer parameters for feature processing. Information fusion is achieved through global attention on the concatenations of queries, keys, and values from each modality. The attention outputs are split for modality-separate fully connected and normalization layers' processing.

research concentrates on a single type of task, like power flow (Hamann et al., 2024) or time-series forecasting (Tu et al., 2025), leading to limited task generalization and applicable scenarios.

Finetuning an open-source language model has demonstrated impressive multi-task capability in power system scenarios (Liu et al., 2025). However, unlike the text-based corpus used by language models, power systems have high-dimensional floating-point state variables, which grow exponentially as the grid size increases. Therefore, simply flattening state variables as text formulations could lead to intensive computation, like what has occurred in flattening pixels in iGPT (Chen et al., 2020). The power system data can be represented in graph structures, where adjacent links refer to transmission line connections and nodal features represent bus-level information, like load consumption, generation, voltages, etc. This property inspires us to design a specially designed encoder, like the visual tokenizer in a vision language model for RGB inputs(Radford et al., 2021; Kim et al., 2024), to efficiently tokenize the graph data. Different from the fixed-shape inputs that vision language models have, the graph data structure could vary as the underlying physical power grids and operational tasks change. Meanwhile, language prompts have been proven to be crucial in instructing multi-modal models to generate specified responses. These require the power system foundation model to not only adapt to graph data with different topologies and task-related nodal features, but also adapt to both language and graph data modalities, which we call *Graph-Language Answering* (GLA), whose workflow is illustrated in Fig.1.

In this paper, we propose a novel architecture design for developing the power system foundation model, called OpenGLA, as demonstrated in Fig.2. In brief, we use graph convolutional networks (GCN) as the core component of the graph encoder, which guarantees topology adaptivity. To achieve task-adaptable node feature processing, we design an adaptive convolution block for nodal feature processing, which takes task-varying nodal feature inputs and generates a fixed-shape embedding for further graph convolution network processing. We project the graph token embedding into the language embedding space. The graph token embedding and language prompt embedding are then fused through global attention in the Mixture of Transformers (MoT) block, which involves separate transformers for each modality (Liang et al., 2024). We also design a transformer detokenizer for the task-varying floating number responses. The model is trained under a supervised training framework to generate correct and precise responses corresponding to each task query.

In summary, our contributions are as follows:

- We propose OpenGLA, a distinct cross-modal ~~power system foundation~~ model designed for adaptivity on diverse system topologies and tasks. Once trained on a unified, multi-

task, and multi-topology dataset, OpenGLA then adapts widely to different operational scenarios on trained grids and tasks.

- We design a practical training strategy that could align the unpretrained graph encoder with the pretrained LLM backbone.

- OpenGLA demonstrates superior performance in the mainstream power system operational tasks and different scales of power grids.

The paper is structured as follows. Section 2 outlines the preliminaries of GCN and cross-modality foundation models. Section 3 presents the detailed designs of OpenGLA, including architecture, training, and instruction designs. Section 4 presents the main experimental results of graph generalization and task adaptivity. Section 5 is the conclusion and the discussion of limitations and future works.

## 2 PRELIMINARIES

### 2.1 GRAPH NEURAL NETWORKS

Graph neural networks (GNNs) generalize deep learning to graph-structured data by propagating and transforming information across node neighborhoods. Early spectral formulations, including ChebNet (Defferrard et al., 2016) and its first-order approximation GCN (Kipf & Welling, 2016), define convolution via graph Laplacian filtering. Given a graph $\mathcal{G} = (\mathcal{V}, \mathcal{E})$ with adjacency matrix $\mathbf{A}$ and feature matrix $\mathbf{X}$, a normalized GCN layer updates node representations as

$$\mathbf{H}^{(l+1)} = \sigma\left(\tilde{\mathbf{D}}^{-1/2}\tilde{\mathbf{A}}\tilde{\mathbf{D}}^{-1/2}\mathbf{H}^{(l)}\mathbf{W}^{(l)}\right), \tag{1}$$

where $\tilde{\mathbf{A}} = \mathbf{A} + \mathbf{I}$ adds self-loops and $\tilde{\mathbf{D}}$ is the corresponding degree matrix. Subsequent spatial message-passing models such as GraphSAGE (Hamilton et al., 2017), GAT (Veličković et al., 2017), and GIN (Xu et al., 2018) further formalize the MPNN framework (Gilmer et al., 2017).

Later work extends GNNs beyond homogeneous graphs: heterogeneous-robust models (Zhu et al., 2020; Chien et al., 2020) decouple ego- and neighbor-information or learn adaptive propagation kernels. In parallel, graph Transformers leverage global attention and positional encodings to overcome locality constraints (Kreuzer et al., 2021). At a larger scale, emerging graph foundation models—such as GraphGPT (Tang et al., 2023)—combine graph encoders with pretrained language models to support instruction following and graph–language reasoning.

### 2.2 MULTI-MODAL FOUNDATION MODELS

Multi-modal foundation models, especially vision-language models (VLMs) leverage both textual and visual data to learn joint representations, facilitating tasks such as image captioning, visual question answering, and cross-modal retrieval. Classic models like CLIP (Contrastive Language–Image Pretraining) (Radford et al., 2021) use contrastive learning to align visual and text representations in a shared embedding space, achieving state-of-the-art zero-shot transfer capabilities. VisualBERT (Li et al., 2019), on the other hand, conditions a language model on visual inputs by mapping them into normal text prefix tokens, enhancing image-to-text generation. Additionally, models leveraging cross-attention mechanisms, such as Flamingo (Alayrac et al., 2022) and BLIP series (Li et al., 2022; 2023; Xue et al., 2024), using vision encoders integrated with text tokenizers, enable fine-grained interactions between modalities, improving contextual understanding. LLaVA series (Liu et al., 2023; 2024b) studies a simpler fusion method, which directly concatenates visual embeddings by projecting them into the language embedding space. These advancements in multi-modal learning have significantly enhanced machine perception, enabling AI systems to bridge the gap between vision and language comprehension.

## 3 OPENGLA

OpenGLA is a cross-modality model that combines graph modality and language modality to efficiently process power system data in graph structures and task instructions in natural languages. In the following section, we introduce the architecture and training designs.

### 3.1 ENABLING MULTI-TASK CAPABILITY USING ADAPTIVE NODAL FEATURE ENCODING

For most of the power system operational tasks, the nodal feature maps could be represented as $X \in \mathbb{R}^{|\mathcal{V}| \times C_{in} \times T}$, where input channel $C_{in}$, length $T$ could vary by tasks, and node number $|\mathcal{V}|$ could vary with topologies. However, it is necessary to learn a mapping $f$ to encode the varying-shape input $X$ as a fixed-shape embedding $H$ for further GCN processing. In this regard, we use the combination of Conv-1d blocks followed by an adaptive pooling layer, as depicted in Fig.2(B), to parameterize the mapping $f$. The convolution blocks are with residual structures to maintain better gradient backpropagation. The adaptive pooling can output fixed embedding $H \in \mathbb{R}^{|\mathcal{V}| \times C_{out} \times F}$ given $C_{out}$ and $F$, whatever the input channel $C$ and length $T$ are. In the mathematical perspective, the mapping is defined as $f : \mathbb{R}^{|\mathcal{V}| \times C_{in} \times T} \to \mathbb{R}^{|\mathcal{V}| \times C_{out} \times F}$.

### 3.2 ENABLING TOPOLOGY ADAPTIVITY USING GRAPH CONVOLUTIONAL NETWORK

We employ the GCN network as the core component of the graph encoder to achieve topology adaptivity as demonstrated in Fig.2(C). GCN's inference is agnostic with the number of nodes and connections, in other words, is independent of the input graph topology. As demonstrated in Eq.1, only the matrix $\mathbf{W}$ is learnable in each GCN layer. Assume the power grid has $|\mathcal{V}|$ nodes and each node has $F$ dimensions of the nodal embedding, then the adjacent matrix is $\mathbf{A} \in \mathbb{R}^{|\mathcal{V}| \times |\mathcal{V}|}$, the hidden state should be $\mathbf{H}^{(l)} \in \mathbb{R}^{|\mathcal{V}| \times F}$ at layer $l$. In order to keep the dimensions of the next-layer feature $\mathbf{H}^{(l+1)}$ remain the same dimension that $\mathbf{H}^{(l+1)} \in \mathbb{R}^{|\mathcal{V}| \times F}$, the learnable matrix should satisfy $\mathbf{W}^{(l)} \in \mathbb{R}^{F \times F}$. This is why GCN can only achieve topology generalization when given fixed nodal feature dimensions. On the other hand, it is also an obstacle to making the GCN encoder task scalable. In addition, we also use residual connections for each GCN layer to maintain better gradient properties.

### 3.3 GRAPH-LANGUAGE MODALITY FUSION THROUGH MIXTURE OF TRANSFORMERS

We conduct the graph-language modality fusion via MoT, inspired by the idea of decoupling non-embedding parameters (including attention projection matrices, feed-forward networks, and layer-norms) according to modalities (Liang et al., 2024). Modality fusion is conducted through the global attention on the concatenations of queries, keys, and values from different modalities, which could be mathematically defined as

$$\text{GlobalAttn}(x, \{\theta_{\text{attn}}^m\}_{m \in \{\text{graph, language}\}}) = \left( \text{softmax}\left( \frac{QK^T}{\sqrt{d_k}} V \right) \right) W_O^{m_i}$$

$$Q = \text{concat}(\{Q_i\}), \quad K = \text{concat}(\{K_i\}), \quad V = \text{concat}(\{V_i\}) \tag{2}$$

$$Q_i = x_i W_Q^{m_i}, \quad K_i = x_i W_K^{m_i}, \quad V_i = x_i W_V^{m_i}$$

where $W_Q^{m_i}, W_K^{m_i}, W_V^{m_i}$ are modality-decoupled projection matrices. We also adopt the pre-norm in Llama(Touvron et al., 2023), then a MoT layer could be defined as:

$$x_i = \text{LayerNorm}_{\text{pre}}^{m_i}(x)$$
$$a = \text{GlobalAttn}\left( x, \{\theta_{\text{attn}}^m\} \right)$$
$$h_i = x_i + a_i \tag{3}$$
$$o_i = h_i + \text{FFN}(\text{LayerNorm}_{\text{post}}^{m_i}(h_i), \theta_{\text{ffn}}^{m_i})$$

$\text{LayerNorm}_{\text{pre/post}}^{m_i}$ are modality-decoupled normalizations. We didn't choose the early fusion method as proposed in LLaVA (Liu et al., 2023; 2024b), which used a dense LLM to fuse concatenated cross-modality token embeddings. MoT significantly lowers the computational cost required to match the performance of dense models. We also prove that MoT could reduce the gradient conflicts and

get better performance in the ablation studies. Another commonly used model sparsity method is Mixture-of-Experts (MoE), where a learned router selectively activates expert networks according to the aspects of input data (Shazeer et al., 2017). However, the routing mechanism could lead to imbalanced expert activations if the router is not properly trained, which further causes inferior performance. MoT resolves this problem through modality-aware sparsity and global attention, removing the routing mechanism and activating transformer parameters according to the input modality.

As mentioned before, we encode graph data using a convolution-based graph encoder that generalizes in both task-varying nodal features and power grid structures. The graph embedding is then projected into the language embedding space with a 2-layer MLP projector as LLaVA-1.5 (Liu et al., 2024b). Then, the graph embedding and language token embedding are fed into the separate graph transformer and language model in MoT, respectively, and fused via the global attention as demonstrated in Eq.2, Eq.3, and Fig.2(D). The fused embedding provides context information to the detokenizer via cross attention to generate expected float responses. For the text responses, we only activate the language model partition in the MoT. We demonstrate two examples of question-answers in Fig.1.

### 3.4 OUTPUT DETOKENIZATION TRANSFORMER

Large language models are not originally designed to generate floating-point outputs. Although some studies on the vision-language-action model have tokenized floating-point numbers through discretization, the language model can generate control vectors for robotic arms (Kim et al., 2024; Brohan et al., 2022; 2023). The key reason for the success of this solution is that the control value range of the robotic arm does not change significantly with the type of task. However, this experience does not apply to power systems. For example, state estimation and optimal power flow, the former is the per-unit value of the output voltage amplitude and phase angle, which are often close to 1 and 0, while the latter is the optimal output of the generator of hundreds of megawatts. If the same discretization is performed on different tasks, there will be huge deviations in the output resolution. In this regard, we designed a lightweight transformer detokenizer for floating-point outputs, which performs cross attention with the hidden state from the last layer of the language model and autoregressively generates the response of each node.

### 3.5 TWO-STAGE TRAINING STRATEGY

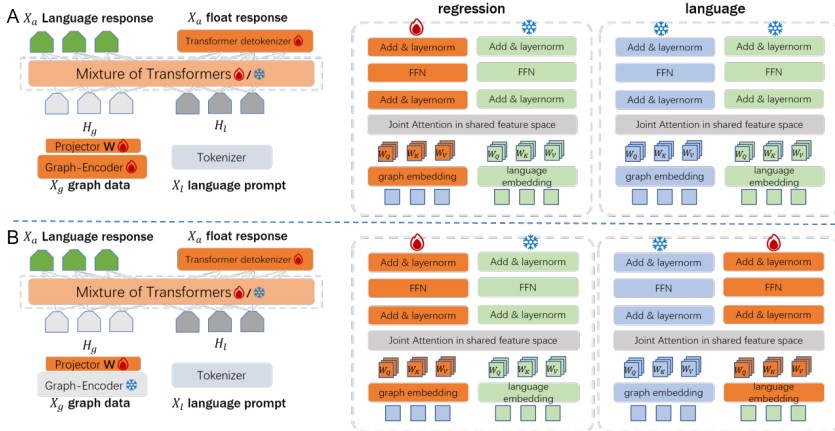

Figure 3: **Two-stage training illustration.** (A) Stage 1. We freeze the parameters of the language model in the MoT no matter what the training objective is, to generate stable embedding tokens of target language responses. For the regression objective, we allow the gradients to flow through the graph transformer parameters. The projector and graph encoder are trained using the cross-entropy loss (language objective) and the MSE loss (regression objective) on embedding token predictions. (B) Stage 2. We freeze the graph-encoder and finetune the projector, language model, and graph transformer using the cross-entropy/mse loss on the autoregressive prediction of generating language/floating responses.

The development of OpenGLA's training strategy was inspired by LLaVA's two-stage training (Liu et al., 2023). This strategy first trains a projector to align the embedding spaces from different

modalities, and then fine-tunes the entire model, excluding the encoder, to generate the desired response. This approach benefits from eliminating the need for separate contrastive learning to align the language embedding space with the non-text embedding space, which typically requires additional, task-irrelevant text annotations. LLaVA's two-stage training, on the other hand, only requires high-quality question-answering datasets from SFT to achieve embedding space alignment.

We follow a similar idea in building OpenGLA's training strategy. In the first stage, as depicted in Fig.3(A), we freeze the language model in MoT and let the gradients flow through not only the projector and graph transformer, but also the graph encoder, since there is no available large-scale pretrained graph data encoder tailored for power systems. We take the language embedding token as labels to train the graph side modules. In the second stage, as demonstrated in Fig.3(B), we freeze the graph encoder and train the projector and the MoT. To be specific, transformer parameters in the MoT are activated according to the response modality. For the regression tasks, the graph transformer is set to be trainable, and the language model is frozen. For the language response, they are reversed. We set the detokenizer to be trainable all the time and use the same dataset in the two stages. More details of the training can be found in Tab.5.

### 3.6 LANGUAGE INSTRUCTION DESIGN

Language instructions are essential for guiding the model to correctly identify the task and the underlying system it is required to address. To this end, we insert a designated placeholder within the instruction, which explicitly indicates the position of the graph data embedding. Furthermore, the instruction specifies that the model should generate appropriate responses conditioned on both the provided graph embedding and the textual instruction. To clarify the overall question–answer structure, we present a detailed specification in Table 6 and illustrative examples in Fig. 1.

## 4 EXPERIMENTS

### 4.1 EXPERIMENTAL SETUP

**Tasks**. 1) Optimal Power Flow (OPF) is the most important optimization task in power system operations, determining the optimal generation $X_a \in \mathbb{R}^{|\mathcal{V}| \times 1}$ given a stack of previous grid states $X_g \in \mathbb{R}^{|\mathcal{V}| \times C_{in} \times T}$, where $C_{in} = 6$ and $T = 1$. 2) Fault detection is a 2-level classification task, determining the type (5 different types of generator, node, line failures) and location (nodes and lines) of the fault that happened, given the grid measurement curves $X_g \in \mathbb{R}^{|\mathcal{V}| \times C_{in} \times T}, C_{in} = 2, T = 240$. It is a language response task as demonstrated in Fig.1. 3) State Estimation determines the true node-level voltage magnitudes and phase angles $X_a \in \mathbb{R}^{|\mathcal{V}| \times 2}$ given the noisy nodal measurements $C_{in} = 3, T = 1$, including active/reactive power and voltage magnitudes. 4) Local Marginal Price (LMP) prediction is a regression task important for the electricity market trading, predicting the next-step LMP given the grid states input. It is with the same output shape as OPF, but its label is marginal cost rather than optimal generator setpoints.

**Grids**. We include different scales of power grids, including IEEE standard cases of 14, 39, 57, 118 and 300 node systems. We also include the Texas 2000 node system to evaluate the scale-up performance in realistic scenarios.

**Dataset**. We acquire open-source load and renewable profiles from Independent System Operators (ISO) and construct 35k scenario cases with different level of load consumption and renewable generation for the OPF task. Then we run ACOPF optimizer provided by Pypower to get the training labels for the OPF and LMP prediction tasks. Meanwhile, we also get the sequential systematic profiles from OPF, including generation and power flow distribution for further transient fault simulations. We consider 5 different fault types, including bus/line fault/outage and generator outage, which could randomly happen. We ensure the data balance of each fault scenarios (including type and locations) in different scales of power grids. We train and evaluate models using separate datasets, with 32k training samples and 3k test samples in each task-grid pair, and 1.05M high-quality QA pairs in total. More details could be found in Appendix A.2.

**Baselines**. For each task, we compare OpenGLA with the vanilla GCN, graph transformer GPS (Rampášek et al., 2022), and finetuned Llama-3.2-1b to evaluate the efficacy of the language modality and graph encoder, respectively. We also include task-specific SoTA baselines, such as

DeepOPF (Pan et al., 2020) and CANOS (Piloto et al., 2024) for OPF, PatchTst (Nie et al., 2022), TimesNet (Wu et al., 2022), and a graph-language model GraphGPT (Tang et al., 2023) for fault detection, GAT (Veličković et al., 2017) and GIN (Xu et al., 2018) for LMP prediction and state estimation to evaluate OpenGLA's final performance.

**Evaluation metrics**. We use widely-accepted metrics to evaluate the fine-tuned model in each task as summarized in Table.8. The OPF pass rate indicates the percentage of power flow convergence when applying the model's OPF responses to the power grid. The optimality gap is the difference of scores between the ground-truth optimizer solutions and model responses, defined following a Lagrangian function that considers economic objectives and operational constraints. The design details of the score function can refer to Appendix.A.6.

**Resources**. All experiments are conducted on a single A100 (80G) GPU machine, demonstrating the resource-efficient design of OpenGLA.

## 4.2 TOPOLOGY AND TASK ADAPTIVE PERFORMANCE

**In-topology scenario generalization**. As demonstrated in Fig.4, OpenGLA outperforms baselines in the separate scenario test dataset among almost all task-grid pairs, demonstrating the efficacy of co-designs of graph data encoding, language modality fusion, and cross-modality output. Pure Llama didn't perform well since it is with the discretization techniques mentioned in (Kim et al., 2024) to achieve float tokenization, and naturally introduce approximation errors. Another reason is that flattening states could disrupt the spatial connections between nodal variables, and the transformer sequence modeling could not effectively reconstruct the spatial dependencies. OpenGLA also outperforms the task-specific models by involving more parameters and language modality, which all significantly enhance the representation capability. Advanced task-specific models like GPS (Rampášek et al., 2022) and GraphGPT (Tang et al., 2023) partly share the design principles and modules with OpenGLA and achieve comparable performance in the corresponding tasks. We also conduct computational efficiency evaluations, whose details could be found in Appendix A.5.

**Robustness to N-1 topology changes**. N-1 is primarily used to evaluate the robustness of the OPF solution to component failures like line outages. In supplementary experiments, we used the output of OpenGLA as the OPF solution and enumerated line failures in subsequent AC power flow simulations. We conducted experiments on the IEEE 14, 39, 57, 118, 300 node systems in the training set, and the results are shown in Table 1.

Table 1: **N-1 robustness evaluations** for baselines and OpenGLA on different grids.

| N-1 OPF pass rate | DeepOPF | CANOS | OpenGLA |
|---|---|---|---|
| IEEE14 | 91.2% | 95.1% | **97.1%** |
| IEEE39 | 80.7% | 83.6% | **85.5%** |
| IEEE57 | 89.5% | 92.4% | **98.7%** |
| IEEE118 | 72.3% | 85.1% | **89.2%** |
| IEEE300 | 76.4% | 87.8% | **92.6%** |

As demonstrated, OpenGLA still maintains a pass rate of over 90% on average, a result that is essentially similar to single-scenario solver methods and significantly outperforms baseline algorithms.

**Generalization on unseen topologies**. We introduced PGLib89 and PGLib500 (Babaeinejadsarookolaee et al., 2019) to evaluate the topology generalization of OpenGLA. We introduced three settings for these three new topologies: zero-shot, transfer learning, and full training. The transfer learning setting uses 1000 training samples per new topology-task pair. Full training uses the full-size 32k samples for each new task-topology pair. As shown in Table 2, we found that zero-shot did not lead to catastrophic performance degradation. The performance drop in the fault localization tasks is due to the model's outputs being absolute bus indexes, which requires some examples to align the bus index space. The performance could be significantly improved through a few samples in transfer learning.

Table 2: **Generalization performance on unseen topologies**. Left/Middle/Right represents zero-shot/1k samples transfer learning/full 32k training, respectively.

| 0-shot/transfer/full | OPF | Fault_type | Fault_loc | State_Est | LMP_Pred |
|---|---|---|---|---|---|
| PGLib89 | 0.241/0.159/0.138 | 0.330/0.182/0.159 | 1.0/0.322/0.217 | 12.871/7.336/4.387 | 4.761/1.574/1.103 |
| PGLib500 | 0.832/0.316/0.274 | 0.389/0.217/0.201 | 1.0/0.524/0.278 | 21.534/11.211/5.138 | 8.203/2.135/1.987 |

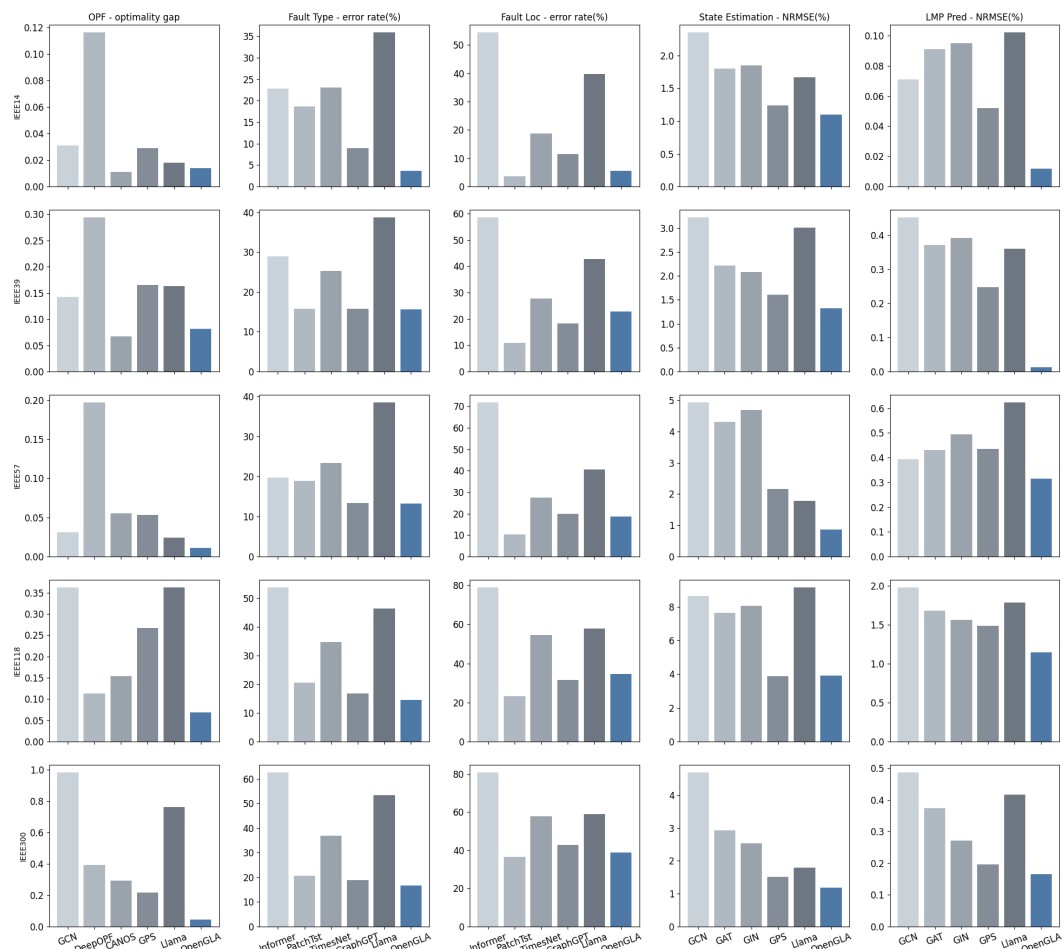

Figure 4: **General performance comparisons** are conducted between OpenGLA and 4 baselines for each task-grid pair, where the lower is better. All models are evaluated using monotonically decreasing metrics. Lower values indicate better performance. All scores are means of 3 runs with different seeds. Our proposed OpenGLA consistently achieves superior results across all tasks and power grids.

### 4.3 SCALING EXPERIMENTS

**Grid Scaling**. We include the Texas2000 system to further evaluate OpenGLA's performance in the larger and more realistic power grid. We didn't include the Llama model as the baseline since pure flattening 2000-node system states leads to GPU memory issues. As presented in Fig.6, OpenGLA achieves better and comparable performance to baselines in all tasks. It is worth mentioning that the reason for the common poor performance of fault localization is that the 2000-node system has more than 22,000 different fault type-location pairs, which results in only a few samples for each type-location pair and leads to limited performance.

**Parameter scaling**. We increase the model capacity to investigate if it helps improve model performance. In the OpenGLA-M(medium), we replace

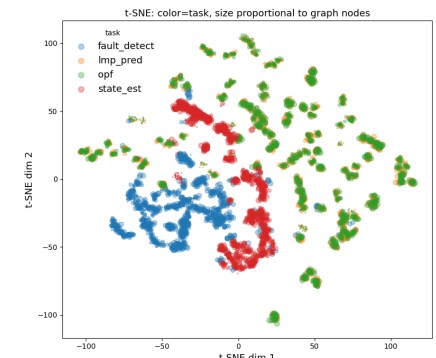

Figure 5: **t-SNE visualizations of learned graph embeddings after stage 1 training**. Colors represent tasks, and node sizes indicate system scales. We use 30 samples for each task-grid pair.

the Llama-3.2-1b with Llama-3.2-3b, and increase the layer number of detokenizer from 2 to 4.

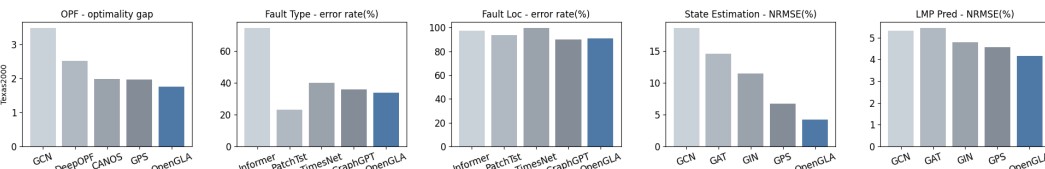

Figure 6: **Scale-up experiment on the large Texas 2000 node system**. All results are mean scores of 3 runs with different seeds. OpenGLA achieves better and comparable performance to task-specific baselines in all tasks.

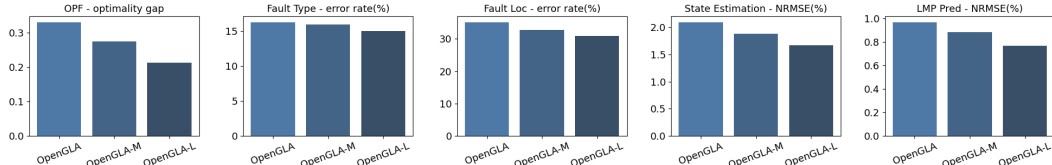

Figure 7: **Parameter scaling investigation**. The performance in each task is the average score on the power grids. All results are mean scores of 3 runs with different seeds.

In OpenGLA-L(large), we use the Llama-3.1-8b, and then the MoT graph transformer is enlarged correspondingly. The layer number of detokenizer in OpenGLA-L is 6. We found that parameter scaling could help improve the model performance.

**Training data proportion**. We also investigate how model performance changes as the proportion of the data used for training changes. We found that in most task-grid pairs, the task performance follows a power law as the training data doubles.

## 4.4 ABLATIONS

**MoT alleviates gradient conflicts**. Gradient conflicts are the main cause that leads to inferior performance in multi-task learning (Yu et al., 2020; Liu et al., 2021). We compare the cosine similarities of different tasks to evaluate the efficacy of the MoT design. As demonstrated in Tab.3, OpenGLA w/ MoT has higher and positive cosine similarities among task gradients, especially between regression tasks and the language response task, where the OpenGLA w/o MoT (dense model with single LLM) contains negative gradient conflicts.

Table 3: **Cosine similarity of gradients of different tasks.** Left represents w/ MoT and right represents w/o MoT.

| (w/ MoT) / (w/o MoT) | Gradient Cosine Similarity ↑ | | | |
|---|---|---|---|---|
| | OPF | LMP Pred | State Est | Fault Detect |
| OPF | - | 0.527/**0.637** | **0.324**/-0.111 | **0.033**/-0.017 |
| LMP Pred | - | - | **0.2336**/0.068 | **0.091**/-0.191 |
| State Est | - | - | - | **0.021**/-0.027 |
| Fault Detect | - | - | - | - |

We also visualized the learned graph embedding distribution, demonstrated in Fig.5. The two-stage training design helps the graph encoder learns high-quality task embedding after stage 1 training. The reason for the overlap in embeddings between LMP prediction and OPF is that they are highly correlated tasks in actual power grid operation (LMP is typically the Lagrangian coefficients of OPF). Furthermore, OpenGLA's graph encoder effectively distinguishes the embeddings of several other types of tasks, which is the basis for achieving strong task performance.

**Single-stage vs. Two-stage training**. We found two-stage training critical as demonstrated in Table 4. Training the randomly initialized graph encoder, detokenizer transformer, and the MoT model simultaneously would corrupt the language model hidden space and cause a catastrophic performance degradation on the fault detection task (language output), as well as significant performance drops in other tasks.

**Number of GCN layers**. We further investigate how changing the number of GCN layers would influence graph embedding learning and the final performance. We evaluate the settings with GCN

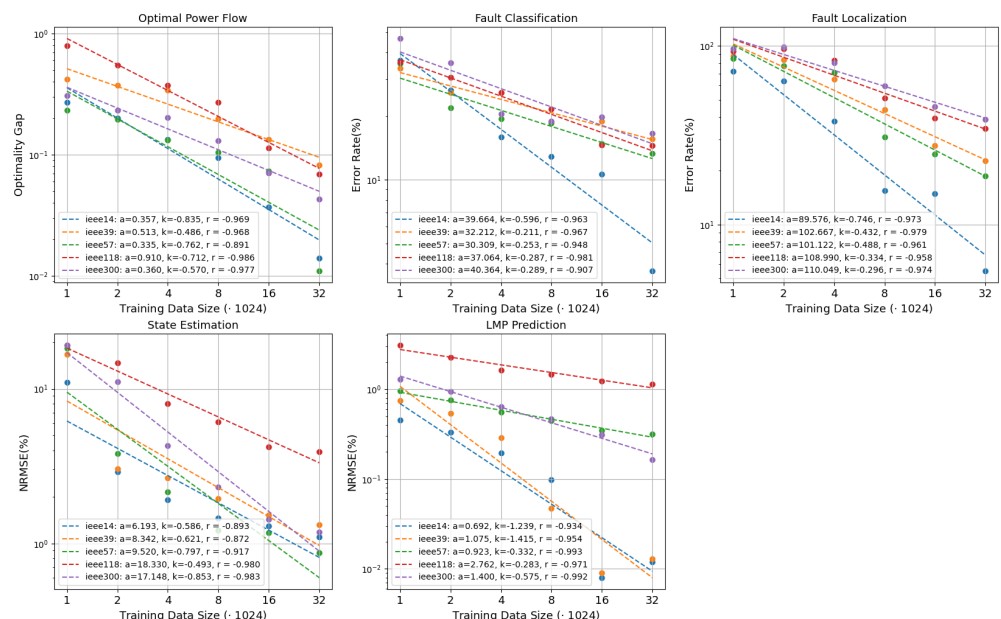

Figure 8: **Power-law scaling of performance with data size**. Five panels report test metrics versus training set size. Each curve represents a different IEEE power-system case ( 14–300 buses); dots are experimental results and dashed lines are power-law fits $y = ax^k$. Across tasks, error decreases approximately as a power law with increasing data.

Table 4: **Single-stage training vs. Two-stage training**. Left represents single-stage training, and the right represents two-stage training.

| Single/Two-phase | OPF | Fault_type | Fault_loc | State Estimation | LMP_pred |
|---|---|---|---|---|---|
| IEEE14 | 0.230/**0.014** | 1.0/**0.037** | 0.870/**0.055** | 105.489/**1.101** | 2.784/**0.012** |
| IEEE39 | 2.561/**0.082** | 1.0/**0.156** | 0.872/**0.228** | 107.395/**1.323** | 1.569/**0.013** |
| IEEE57 | 0.416/**0.011** | 1.0/**0.133** | 0.869/**0.187** | 104.437/**0.875** | 2.752/**0.315** |
| IEEE118 | 0.294/0.069 | 1.0/**0.145** | 0.816/**0.346** | 76.878/**3.921** | 11.068/**1.142** |
| IEEE300 | 1.102/**0.043** | 1.0/**0.166** | 0.890/**0.388** | 100.907/**1.186** | 3.950/**0.165** |

layer numbers 1, 3, 5, 7 on IEEE14, 39, 57, 118, and 300 node systems. As demonstrated in Fig.12, the setting of 3 GCN layer outperforms the other settings, indicating that insufficient or excessive GCN layers both affect graph embedding learning and the performance on downstream tasks.

**Qualitative visualization**. We also provide a qualitative visualization of the fault localization process on the IEEE14 system, as demonstrated in Fig.13.

## 5 CONCLUSION & DISCUSSION

In this paper, we present the OpenGLA model, a novel architecture ~~for power system foundation model, by offering a robust architecture design~~ that combines graph-based representations with natural language processing for complex tasks. The model's architecture effectively addresses the challenges posed by the varying topologies and operational scenarios of power grids, demonstrating remarkable adaptability and scalability.

It remains to explore whether the performance could be further improved by extremely scaling model parameters and dataset scales. Meanwhile, it is also an open question if the zero-shot/few-shot performance on unseen topologies and tasks could be improved by scaling training and data. Finally, the computational efficiency could be further improved by proposing a better design on the model output side, rather than the autoregressive generation we currently use. This model is generalizable towards other graph-related tasks, such as urban transportation network operations.

**Ethics Statement.** This work concerns power-system operation data and simulations. We use only synthetic or publicly available data; no personally identifiable information is involved. We discuss potential risks of misuse (e.g., unsafe operating suggestions) and provide safeguards: (1)

we release code and data for research replication only under an appropriate license; (2) our model outputs are not intended for real-time grid control without a certified operator-in-the-loop; (3) we document dataset generation, assumptions, and known limitations; (4) we report failure cases and out-of-distribution behaviors in the appendix. We comply with privacy, security, and legal requirements for all data sources and releases.

**Reproducibility Statement.** We release an anonymized zip file with training and evaluation scripts and configuration files. Hyperparameters are listed in Tab.5; dataset creation, preprocessing, and splits are detailed in Appendix A.2-A.6; model and training details are in Appendix A.3. We repeated each training 3 times using different seeds and report the mean scores. We also report hardware (A100-80GB) and time usage in Appendix A.5. It is also recommended to have a GPU machine with at least 8 CPUs and more than 200G memory. All data for training and evaluations is accessible via an anonymous link `https://dataverse.harvard.edu/previewurl.xhtml?token=5f95c287-4c0e-4c05-a970-411b4ac2c7a4`. Publicly available code will be accessible upon paper acceptance.

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

## A APPENDIX

### A.1 LLM USAGE

We used an LLM solely for grammar editing and wording suggestions after the research, methods, and results were completed. The model did not generate technical content, ideas, code, analyses, or experiments. Authors take full responsibility for all contents.

### A.2 DATASET PREPARATION

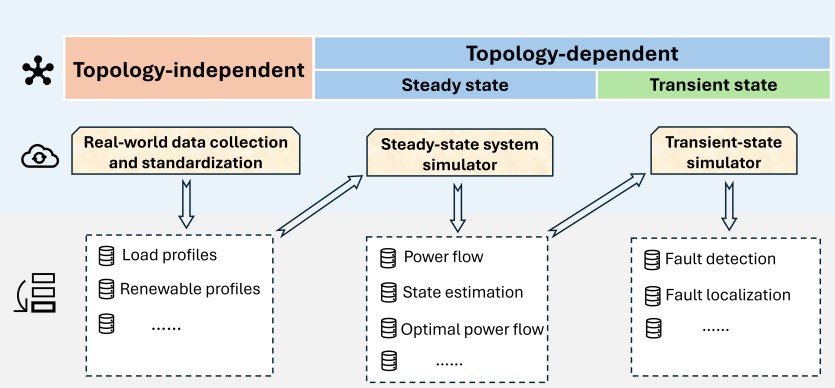

Figure 9: **Data Generation Flow**. We introduce a flexible data generation pipeline that enables easy extension and further expansion of the dataset.

Power system dynamics are typically divided into different time resolutions for research due to the resolution-varying physical characteristics of connected devices. There are specific dynamic

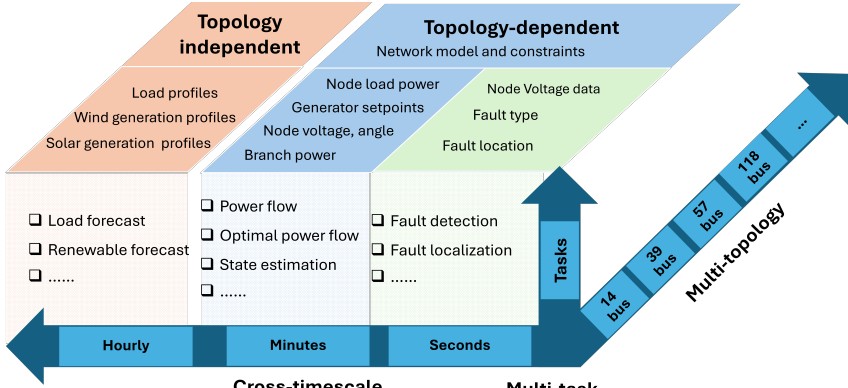

Figure 10: **Dataset Structure.** We include three key data dimensions: timescales, topologies and tasks, along with additional topology-independent data (e.g., renewable and load profiles). Topology-independent data can be applied to each topology as standardized system inputs.

modeling methods and simulation software for each time scale. In this paper, we propose a hierarchical data generation flow as depicted in Fig.9. It utilizes publicly available, normalized load and renewable time-series profiles - unrelated to the underlying dynamics of power grids (topology-independent) - to create scenarios for power flow simulations. Then, steady-state (minute-level physics behavior) simulations generate systematic states given various scenarios using the iteration of optimal power flow and power flow simulation. The transient-state (milliseconds to seconds-level) physics behavior uses steady-state profiles as the start scenarios, generating transient voltage, current curves, and frequency changes by simulating diverse faults happening at possible devices, which are highly coupled with topology and system parameters, and are thus called topology-dependent. To be specific, the data generation pipeline consists of three major steps:

- **Time-series data processing.** We collect high-resolution temporal profiles for load consumption and renewable generation using historical data and high-fidelity synthetic profiles across major ISO territories in the U.S. These time-series profile data (referred to as Topology-independent data) will serve as universal inputs to subsequent simulations that are topology-dependent.

- **Steady-state simulation.** These topology-independent data are then aligned and used as scenarios for power flow simulations to generate systematic profiles. We use Pypower (Zimmerman, 2011; Zimmerman et al., 2010), a popular open-source Python-based tool for steady-state modeling, to perform the power flow simulations, which support key power system operations such as power flow analysis. We also use Pypower to perform optimal power flow optimization and generate labels. Each scenario corresponds to a timestamp and contains full system information such as nodal voltages, power generation, and line power flow.

- **Transient-state simulation.** Transient-state simulations are conducted given the steady-state simulation outputs as the initial conditions on selected network topologies. We use PowerWorld Simulator (Thayer et al., 2020), a widely adopted tool in the power system industry and academia, to simulate transient responses. Synthetic faults are introduced to generate system dynamic behaviors, including voltage magnitude, phases, frequency curves, etc. These topology-dependent data can be further used for downstream tasks such as fault classification and dynamic behavior prediction.

The modularity of this flow makes it scalable with new topologies or time-series drivers without modifying the simulation architecture. The dataset currently includes multiple grids (i.e., IEEE 14, 39, 57, 118, 300 and Texas2000-bus systems) and supports from seconds to hourly resolutions of dynamic behaviors.

Then we can generate training samples for the fault detection task using the transient voltage, current and frequency curves as the input data, and using the simulated faults as ground-truth labels. Things are similar in OPF, state estimation, and lmp prediction. We generate samples using a pair of simulated system variables and the optimizer's outputs/masked system information.

## A.3 TRAINING DETAILS AND HYPERPARAMETERS

Details of hyper-parameters could be referred to Tab.5.

Table 5: Training hyperparameters.

| Hyperparameter | Description | Value |
|---|---|---|
| Optimizer | AdamW | |
| Learning rate | Stage 1 LR | 2e-4 |
| Learning rate | Stage 2 LR | 2e-5 |
| Batch size | update batch size | 1 |
| Gradient accumulation | gradient accumulation steps | 4 |
| Epochs | Number of training epochs | 2 |
| Weight decay | L2 regularization coefficient | 0.01 |
| Dropout | Drop probability | 0.1 |
| Scheduler | Cosine decay w/ warmup | |
| Warmup steps | Steps before decay | 1000 |
| Grad clip | Global norm limit | 1.0 |
| Hidden dim | Graph embedding dim | 1024 |
| Hidden dim | Language embedding dim | 1024 |

## A.4 LANGUAGE INSTRUCTION DESIGNS

The details of language instructions are demonstrated in Tab.6.

| Task | Instruction | Response |
|---|---|---|
| OPF | [graph_tokens] This is an operation scenario of {network} bus system. What is the best active power setpoint of generators? | A $\mathbb{R}^{|\mathcal{V}|\times 1}$ vector of generator setpoints. |
| Fault Detection | [graph_tokens] These are fault nodal voltage curves in {network} bus system. What are the fault type and fault location? | The fault type is {fault_type}, occurred at bus {fault_bus} (between bus {fault_bus1} and {fault_bus2}). |
| State Estimation | [graph_tokens] This is measurements of voltage magnitudes, active power injection and reactive power injection in {network} bus system. What are the real states of voltage magnitude and phase angles? | An $\mathbb{R}^{|\mathcal{V}|\times 2}$ array of estimated voltage magnitudes and phase angles. |
| LMP Prediction | This is an operation scenario in {network} bus system. What is the locational marginal price? | A $\mathbb{R}^{|\mathcal{V}|\times 1}$ vector of nodal marginal prices. |

Table 6: Detailed question-answer designs for each task.

## A.5 INFERENCE & TRAINING EFFICIENCY

We evaluate the computational efficiency of the proposed model across different power grids as demonstrated in Fig.11 and Tab.7. We pick GCN and Informer as the representatives of the task-specific models for better illustrations, since there are no significant differences in their inference throughput. Intuitively, task-specific models have the highest effiency since their inference computation could be completely parallelized. OpenGLA achieves a trade-off between the pure autoregressive model and task-specifics by using the graph encoder to reduce the sequence length of the input grid state from full expansion to the number of nodes. In addition, OpenGLA also achieves 6 times less training time compared to the full flattening way for finetuning a pure LLM. We didn't compare OpenGLA with conventional models since they are trained task-specifically.

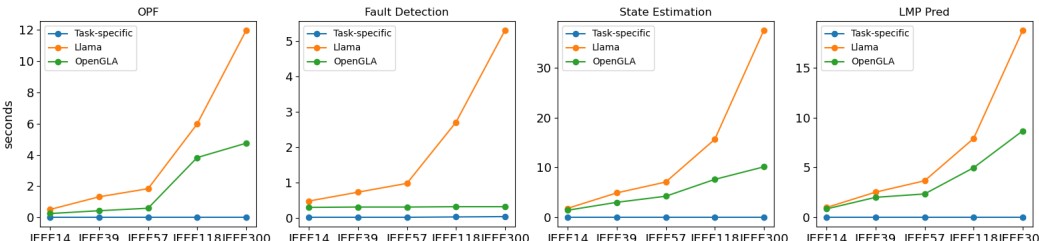

Figure 11: **Inference time comparisons** are conducted for the three categories—Specific (blue, including GCN and Informer), Llama-3.2-1b (orange, pure language input), and OpenGLA (green)—across four tasks: OPF, Fault Location, State Estimation, and LMP Prediction, evaluated on five power grids: IEEE14, IEEE39, IEEE57, IEEE118, and IEEE300. All experiments are conducted on a single A100 (80G) GPU machine.

|  | Pure LLM | OpenGLA |
| --- | --- | --- |
| Total training Time (h) | 190 | 32 |

Table 7: **Training time comparisons** are conducted for the two categories, pure LLM (Llama-3.2-1b, all flattening), and OpenGLA (with graph encoder)—across four tasks: OPF, Fault Location, State Estimation, and LMP Prediction, evaluated on five power grids: IEEE14, IEEE39, IEEE57, IEEE118, and IEEE300. All experiments are conducted on a single A100 (80G) GPU machine.

## A.6   EVALUATION METRICS DETAILS

| Task | Metric design |
| --- | --- |
| OPF | Optimality gap |
| Fault Detection | Error rates of type prediction and localization(%) |
| State estimation | NRMSE(%) |
| LMP prediction | NRMSE(%) |

Table 8: Evaluation metric designs for each task.

The metric of optimal power flow response is designed as a weighted average of the following components, considering the power system steady-state operational constraints and economic objectives.

**Line overflow.** To ensure the security of power transmission, we establish a reward mechanism that takes into account the line loadings. The current load rate is defined as the ratio of the transmission current to the maximum transmission current, which is limited by the thermal capacity of the line. To prevent line congestion and outages, it is desirable for the transmission lines to maintain a reasonable load level. Consequently, we design the reward function for this aspect as follows:

$$r_{\text{overflow}} = 1 - \frac{\sum_i \min(\rho_i, 1)}{n_{\text{line}}} \quad (4)$$

where $\rho_i$ indicates the current load rate of line $i$. $n_{\text{line}}$ represents the line number.

**Renewable consumption.** To optimize the utilization of renewable energy, a reward function is formulated to encourage the scheduling agent to consume more renewable energy. This reward function is based on the renewable energy consumption rate, which is defined as the ratio of the total power currently consumed from renewable sources to the maximum power generated by renewable sources. A higher renewable energy consumption rate corresponds to a higher reward, thereby incentivizing the agent to maximize the use of renewable energy. This reward is designed as follows:

$$r_{\text{renewable}} = \frac{\sum_i \text{p}_i}{\sum_i \text{p}_i^{\max}}, \quad i \in \text{renewable units} \quad (5)$$

where $\text{p}_i$ represents the power output of generator $i$, $\text{p}_i^{\max}$ indicates maximal power generation capability of renewable generator $i$. The closer $\text{p}_i$ and $\text{p}_i^{\max}$ are, the higher the renewable consumption reward.

**Power balancing.** The balanced generator aims to balance the residual power and eliminates discrepancies between power generation and load consumption, while its generation capability is limited. If its power output exceeds its operational boundaries, a power mismatch occurs, which may

result in load shedding or blackouts as previously mentioned. To prevent such mismatches and maintain safe operation, we design the reward function as follows:

$$r_{\text{balance}} = -\left( \frac{\max(p_{\text{bal}} - \overline{p_{\text{bal}}}, 0)}{\overline{p_{\text{bal}}} - \underline{p_{\text{bal}}}} + \frac{\max(\underline{p_{\text{bal}}} - p_{\text{bal}}, 0)}{\overline{p_{\text{bal}}} - \underline{p_{\text{bal}}}} \right) \tag{6}$$

where $\overline{p_{\text{bal}}}$ and $\underline{p_{\text{bal}}}$ indicate the upper bound and the lower bound of the balanced generator's active power per step. If the balanced power $p_{\text{bal}}$ is out of bounds, there would be a penalty.

**Operating cost.** We formulate a reward function for the operational costs of thermal units while considering the negligible costs of renewable energy generation. Specifically, the operating costs of thermal units are represented as quadratic functions of output power, and additional costs are incurred for the startup/shutdowns of thermal units. As for renewable sources, their operating costs are considered to be negligible as they do not rely on fossil fuels for power production. The operating cost reward is designed as follows:

$$r_{\text{cost}} = -\frac{\sum_i c_{i,2} p_i^2 + c_{i,1} p_i + c_{i,0} + \mathcal{I}(s_i, s_i^-) c_{\text{on-off,i}}}{Z} \tag{7}$$

where $c_{i,2}, c_{i,1}$ and $c_{i,0}$ are the second order, first order and constant coefficients of the operation cost of generator $i$, respectively. The coefficients of renewable units are much lower than that of thermal units. $p_i$ represents the power output of generator $i$. $s_i$ represents the on-off status of generator $i$, and the $s_i^-$ is the status 1-step advance. $c_{\text{on-off,i}}$ is the startup and shutdown costs of generator $i$. $\mathcal{I}(s_i, s_i^-)$ is an indicative function that turns to be 1 if $s_i \neq s_i^-$, otherwise 0. $Z$ is the normalization factor set as $10^5$ in experiments.

**Reactive power.** Reactive power plays a vital role in supporting the voltage stability of the power grid. However, the reactive power output capacity of the generators is constrained. While exceeding this limit is not catastrophic, excessive reactive power compensation can significantly increase operational cost. In light of these considerations, we design the reactive power reward as follows:

$$r_{\text{reactive}} = \exp\left( -\sum_i \left[ \frac{\max(q_i - \overline{q_i}, 0)}{\overline{q_i} - \underline{q_i}} + \frac{\max(\underline{q_i} - q_i, 0)}{\overline{q_i} - \underline{q_i}} \right] \right) - 1 \tag{8}$$

where $q_i$ is the reactive power of generator $i$, and $\overline{q_i}, \underline{q_i}$ are the upper bound and the lower bound of generator $i$. There would be a penalty if any generator violates its reactive power constraint.

**Bus voltage.** In power system operation, it is common practice to limit node voltage magnitudes within the range of 0.95-1.05 per unit. If the voltage magnitude at a node is too low, it can result in a significant increase in the transmission loss of the grid. Conversely, if the node voltage magnitude is too high, it requires more reactive power compensation and may cause the generator's reactive power to exceed its upper limit. To regulate the node voltage magnitudes within specified ranges, we design the bus voltage reward similarly to the reactive power reward.

$$r_{\text{voltage}} = \exp\left( -\sum_i \left[ \frac{\max(v_i - \overline{v_i}, 0)}{\overline{v_i} - \underline{v_i}} + \frac{\max(\underline{v_i} - v_i, 0)}{\overline{v_i} - \underline{v_i}} \right] \right) - 1 \tag{9}$$

where $v_i$ is the voltage magnitude of bus $i$, and $\overline{v_i}, \underline{v_i}$ are the upper bound and the lower bound of voltage magnitude of bus $i$. There would be a penalty if any bus violates its voltage magnitude constraint.

## A.7    SUPPLEMENTARY ABLATION RESULTS

**Changing the number of GCN layers** is demonstrated in Fig.12.

**Qualitative visualization of fault localization process** is demonstrated in Fig.13.

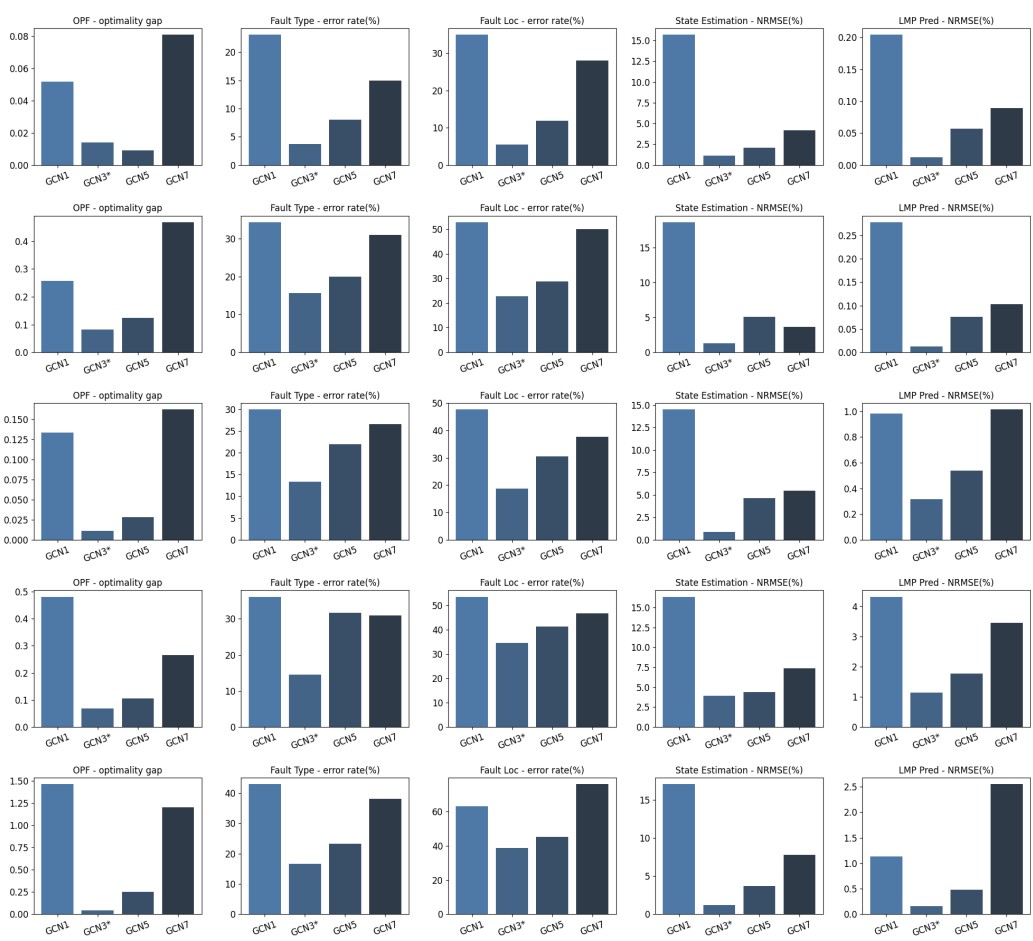

Figure 12: **Ablation on number of GCN layers**. As we can see, the setting of 3 GCN layers achieves empirically better performance on different task and topology pairs. A shallow graph head may affect the expression capability of graph embedding learning. However, too many GCN layers may cause smoothed graph embedding and affect down-stream performance.

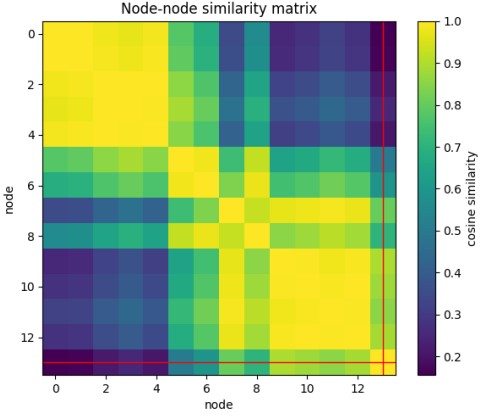

(a) Heatmap of node-node graph embedding similarity

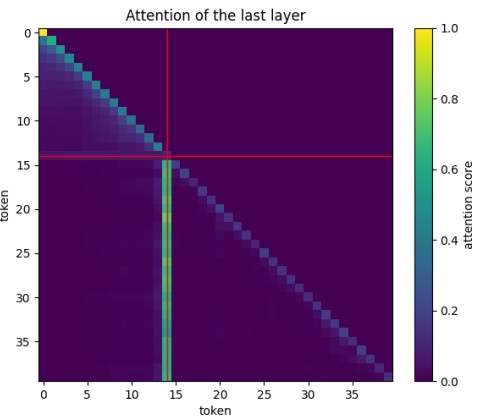

(b) Heatmap of last-layer attention

Figure 13: **Visualization of fault localization process**. There is a bus strip fault that occurred at bus 14 in the IEEE14 system. As observed in the left figure, the 14th-node embedding is not strongly correlated with other nodes', indicating the extremely abnormal features with the fault. Heatmap of the last-layer attention is visualized in the right figure, the language backbone successfully captures the differences in graph embedding and assigns the highest attention weight on the 14th-node graph token. (The first 14 are graph tokens, and the rest are language prompt tokens.)

