# OpenReview forum: "OpenGLA: Topology and Task Adaptive Foundation Model for Power System Graph-Language Answering"
_ICLR.cc/2026/Conference — Submitted to ICLR 2026_

### Official Review · Reviewer_KGmd · 2025-10-30

**Soundness:** 3
**Presentation:** 3
**Contribution:** 3
**Rating:** 6
**Confidence:** 3

**Summary:**

This paper introduces OpenGLA, a novel foundation model specifically designed for Power System Graph-Language Answering (GLA) tasks. The authors argue that applying existing Large Language Models (LLMs) directly to power systems is challenging due to the inherent graph-structured nature of the data and the strict requirements for high-precision, floating-point outputs.

To address this gap, OpenGLA introduces an innovative multi-modal architecture with the following core contributions:

1. Graph Encoder: It utilizes a Graph Convolutional Network (GCN) to effectively encode grid data with varying topologies, achieving adaptability to different power system structures.
2. Mixture-of-Transformers (MoT): A sparse model designed to efficiently fuse graph embeddings with natural language instructions. It uses separate parameters for each modality, which effectively mitigates gradient conflicts in multi-task learning.
3. Specialized Output Module: To handle the need for precise floating-point outputs, the model incorporates a lightweight transformer "detokenizer" that generates numerical vectors instead of discrete text tokens.
4. Two-Stage Training Strategy: Inspired by vision-language model training, the strategy first aligns the feature space of the graph encoder with the pre-trained LLM, followed by end-to-end fine-tuning.

The authors conduct comprehensive experiments across four critical power system tasks: Optimal Power Flow (OPF), Fault Detection, State Estimation, and Local Marginal Price (LMP) prediction. These experiments span multiple grid scales, from the IEEE 14-bus system to the realistic Texas 2000-bus system. The results demonstrate that OpenGLA significantly outperforms existing task-specific models and fine-tuned LLMs in nearly all tested scenarios, showcasing its superior performance, generalization, and scalability.

**Strengths:**

1. Clear Problem Formulation with High Application Value: The paper is well-motivated, precisely targeting the challenges of applying foundation models to the critical domain of power systems. The proposed solution has significant real-world relevance and substantial potential for industrial application.
2. Innovative and Well-Justified Model Architecture: The design of OpenGLA is both innovative and thoughtfully crafted. Each component addresses a specific and significant challenge: the GCN for topological adaptability, the adaptive nodal encoder for task-input flexibility, the Mixture-of-Transformers (MoT) for efficient multi-modal fusion, and the Transformer Detokenizer for handling precise numerical outputs. Together, they form a cohesive and powerful system.
3. Extensive and Compelling Experimental Validation: One of the most outstanding merits of this paper is its rigorous and comprehensive experimental validation. The authors have tested their model across a diverse set of tasks and on power grids of vastly different scales (from standard IEEE cases to the large-scale Texas 2000-bus system). The comparisons against multiple strong baselines provide irrefutable evidence of the model's superiority. The successful evaluation on the Texas grid, in particular, demonstrates the model's potential to scale to practical, large-scale applications.
4. Demonstrates Excellent Scalability: The paper showcases a key characteristic of a true foundation model. Through the scaling law analysis (Figure 8), it shows that the model's performance improves predictably with increases in parameter and data scale. This provides strong evidence for its future potential and aligns the work with the principles of large-scale model development.

**Weaknesses:**

1. Limited Evaluation of Topological Generalization: The most significant limitation is that the model's generalization capabilities are not tested in the strictest sense. According to the experimental setup, the model is trained and evaluated on the same grid topology (e.g., trained on IEEE-118 data, tested on IEEE-118 data). For a model positioned as a "foundation model," a more rigorous test would be cross-topology generalization—for instance, training on a set of topologies (e.g., 14, 39, and 57-bus systems) and evaluating its zero-shot performance on an entirely unseen topology (e.g., the 118-bus system). The current evaluation primarily demonstrates generalization to new operational scenarios within a known structure, rather than to new structures themselves.
2. Performance Trade-offs Against Specialized Models: As observed in Figure 4, while OpenGLA's overall performance is exceptionally strong, task-specific SOTA models like PatchTST can achieve comparable or even slightly better performance on their niche tasks (i.e., time-series-based fault detection and localization). This is not a flaw but rather highlights a classic trade-off between a generalist and a specialist. The paper could benefit from a more direct discussion of this trade-off, acknowledging that while OpenGLA’s strength is its unified and versatile nature, highly optimized specialist models may still hold an edge in their specific domains.
3. Absence of Hyperparameter Sensitivity Analysis: The appendix provides the final set of hyperparameters used for training, but the paper lacks a sensitivity analysis. A study on how performance is affected by key hyperparameters (e.g., learning rate, number of GCN layers, or embedding dimensions) would significantly strengthen the paper's claims of robustness. Without it, it is difficult to assess the model's stability and how easily the results can be reproduced.
4. Unfulfilled Promise of Physical Interpretability: The ABSTRACT rightfully highlights "physical interpretability" as a critical requirement for power system applications, setting it up as a key motivation for the proposed model. However, the paper fails to deliver on this promise. The experimental section focuses entirely on performance metrics like accuracy and error rates, without providing any analysis to verify that the model's reasoning aligns with the physical laws of power systems. There is no qualitative analysis, such as visualizing conducting case studies, to show why the model makes its predictions. This omission creates a significant gap in the paper's narrative, leaving the claims of suitability for critical infrastructure partially unsubstantiated.
5. Inadequate Benchmarking Against Non-Learning Baselines: For operational tasks like Optimal Power Flow (OPF), a key motivation for adopting a deep learning model is the potential for substantial inference speedup over traditional numerical solvers. However, the paper's efficiency evaluation is confined to comparisons with other learning-based models. It critically omits any performance benchmark against standard, industry-accepted numerical solvers (e.g., the interior-point or simplex methods available in PyPower/MATPOWER). This is a significant flaw, as it fails to demonstrate the practical utility of the proposed method. Without this essential comparison, it is impossible to assess whether OpenGLA provides a meaningful speed advantage that would justify its use over conventional, highly reliable methods. Consequently, the paper's claims regarding computational efficiency and practical viability are not sufficiently substantiated.
6. Limited Scope of Ablation Studies: The paper provides a valuable ablation study demonstrating the effectiveness of the Mixture-of-Transformers (MoT) module in mitigating gradient conflicts. However, the analysis does not extend to other novel components of the OpenGLA architecture. For instance, the adaptive nodal feature encoder (Figure 2B) employs a specific 1D-convolutional design to handle task-varying inputs. It would be insightful to see a comparison against a simpler alternative, such as a standard Multi-Layer Perceptron (MLP), to quantify the actual benefits of this specific design choice. Without a more comprehensive ablation, it is difficult to ascertain whether the complexity of each proposed component is fully justified, or if a simpler design could have yielded comparable performance.

**Questions:**

1. On Cross-Topology Generalization: Regarding the model's adaptability, the current experiments demonstrate strong generalization to new scenarios within previously seen topologies. To truly substantiate the claims of OpenGLA as a "foundation model," have the authors considered conducting a zero-shot cross-topology generalization test? For example, what is the performance if the model is trained on the IEEE 14, 39, and 57-bus systems and then evaluated directly on the unseen IEEE 118-bus system? Any insight into this more challenging scenario would be highly valuable.
2. On the Necessity of the Mixture-of-Transformers (MoT): The paper convincingly shows that MoT alleviates gradient conflicts compared to a dense model. However, a simpler baseline for multi-modal fusion, such as the one used in LLaVA (projecting visual tokens and concatenating them with language tokens), was not included in this specific comparison. Could the authors comment on why the more complex MoT architecture was chosen over this simpler, widely-used concatenation approach, and what specific advantages it provides in the context of graph-language fusion?
3. On Physical Interpretability: The ABSTRACT rightly emphasizes physical interpretability as a key requirement for power system models, yet the experimental validation focuses on quantitative metrics. Could the authors provide a qualitative case study to shed light on the model's decision-making process? For instance, in a fault localization task, is it possible to visualize the attention weights of the Mixture-of-Transformers to verify that the model correctly focuses on the grid areas physically close to the fault? Such an analysis would be crucial for building trust and ensuring the model's reasoning is physically grounded.

---

> ### Author Response · Authors · 2025-11-21
>
> We sincerely thank the reviewer for careful reading of our paper and insightful comments. The following are our responses.
>
> **Topology generalization**. This is a common concern, and please refer to the general comments.
>
> **Hyperparameter sensitivity analysis**. We add ablation studies on the number of GCN layers, as well as a comparison between single-stage training and two-stage training. In one word, single-stage training causes catastrophic performance degradation. We will update the ablation result of changing the GCN layer number as soon as it is available.
>
> **Necessity of MoT**. Actually, the w/o MoT setting in the MoT ablation is exactly the Llava-style setting. As demonstrated in the Table. 1, w/o MoT leads to more gradient conflicts and also inferior performance.
>
> **Physical Interpretability**.  We will try to visualize the eigenvalues of graph embedding in the fault localization tasks in the final revision. However, for the regression tasks, it is difficult to appropriately visualize the model behavior.
>
> We hope our responses could address your concerns.

---

> > ### Comment · Reviewer_KGmd · 2025-11-26
> >
> > Thanks for your response. I appreciate that the authors have conducted a cross-topology generalization test and evaluated OpenGLA on two unseen grid topologies. However, the noticeable degradation in zero-shot performance on PGLib500 raises concerns about the claimed robustness. The author's explanation in "General response for N-1 and unseen topologies " that the performance drop in fault localization is caused by bus-index misalignment is reasonable but may not fully account for the large degradation observed (e.g., fault-loc from 0.278 to 1.0 in PGLib500). This means the model almost completely fails to correctly identify fault locations without any adaptation to the new topology.  The "full-training" results mainly reflect in-topology fitting rather than unseen-topology generalization. Therefore, the robustness claim should rely primarily on zero-shot and transfer-learning outcomes. Additional evidence, such as error distribution or visualization, would help support this claim.
> >
> > Besides, for *Hyperparameter sensitivity analysis*, I could not locate the reported ablation studies on the number of GCN layers in the revised manuscript. Please ensure that the experiments are actually included and clearly highlighted to help reviewers locate them.

---

> > > ### Author Response · Authors · 2025-11-27
> > >
> > > Thanks for the reviewer's timely response. The adaptability we claimed in our paper refers to the trained topologies and tasks. Given that power systems are critical infrastructure, practitioners may be less willing to take the risk of zero-shot topology generalization, and instead value the ability to perform cross-task and topology ability after one-time training (thus avoiding arduous case-by-case development). Considering the risk appetite of the power industry, transfer learning and full-data fine-tuning on new topologies and tasks are necessary. We provide transfer and full-training results precisely to demonstrate our model's transferability in this scenario. Full training also serves as an upper bound for the zero-shot and transfer learning settings.
> > >
> > > The ablation experiments for the number of GCN layers are shown in Fig. 12 of the latest PDF revision. We have also highlighted other modifications in blue. In general, insufficient (affecting representational power) and excessive GCN layers (affecting embedding smoothness) both negatively impact the performance of downstream tasks. Three layers are the optimal empirical choice in our experiments.
> > >
> > > Fig.13 is a qualitative visualization of the fault localization process on the IEEE14 bus system. As demonstrated, the graph embedding learning can capture the abnormal feature characteristics of the fault bus. The language backbone can also capture the differencies in nodal embedding and assign the highest attention weight on the fault node token.

---

### Official Review · Reviewer_JsEo · 2025-10-31

**Soundness:** 2
**Presentation:** 2
**Contribution:** 2
**Rating:** 2
**Confidence:** 4

**Summary:**

This paper proposes OpenGLA, a graph-language foundation model for power system operations that combines a topology-adaptive GCN encoder with language instructions to handle diverse tasks (OPF, fault detection, state estimation, LMP prediction). The model uses an adaptive Conv-1D nodal feature encoder to handle task-varying inputs, projects graph embeddings into language space, fuses modalities via Mixture-of-Transformers (MoT), and employs a lightweight transformer detokenizer for floating-point outputs. The model is trained on synthetic data covering multiple IEEE test cases and evaluated across different tasks and grid topologies, demonstrating superior performance compared to task-specific baselines.

**Strengths:**

Domain-Motivated Design: The paper addresses a real need in power systems by designing topology-adaptive components (adaptive Conv-1D encoder + GCN) and a transformer detokenizer for floating-point outputs, which are well-motivated by domain requirements rather than just applying off-the-shelf models.

Evaluation: The experimental evaluation spans multiple tasks (OPF, fault detection, state estimation, LMP prediction) and grid scales (IEEE 14-300 buses, Texas2000),  ablation studies show the efficacy of the MoT design in reducing gradient conflicts.

Task-Specific Performance: OpenGLA achieves relative good performance compared to task-specific baselines across most task-grid pairs, with detailed metrics and statistical reporting.

**Weaknesses:**

Overstated Foundation Model Claims: The paper claims to present a "foundation model" but provides no evidence of the key properties that define foundation models: (1) no large-scale pretraining on diverse data, (2) no zero-shot or few-shot generalization to new tasks/topologies, (3) training is done separately for each task-grid pair with only 32k samples each, (4) all evaluation is within-distribution on synthetic data. This is essentially a multi-task model for power systems, not a foundation model in the established sense.

Insufficient Literature: There is no related work section. It seems authors have limited understanding of the broader graph learning literature. In the preliminary, the paper positions GCN (2016) as the core GNN method while ignoring the evolution of graph neural networks, spectral methods, heterophilic GNNs, and recent graph foundation models (e.g., GraphGPT, G-Retriever, InstructGLM mentioned in the first paper's review). The comparison is limited to outdated baselines (GAT, GIN from 2017-2018) rather than recent graph transformers or graph-language models.

Scalability: Despite targeting "critical infrastructure," all experiments use synthetic data from simulators (PyPower, PowerWorld). The largest system (Texas2000) shows poor performance on fault localization (Fig. 5), and there's no evaluation on real SCADA/PMU data from actual grids. For a system claiming to support operational decision-making, this is a critical gap affecting credibility and practical applicability.

Limited Architectural Novelty: The core architecture is largely assembled from existing components: GCN (Kipf & Welling 2016), standard projector (from LLaVA), MoT (Liang et al. 2024), and LLaMA backbone. The adaptive Conv-1D encoder is a straightforward application of adaptive pooling. The main contribution is domain-specific engineering rather than fundamental methodological innovation.
Weak Generalization Evidence: The paper provides no evaluation of generalization to: (1) unseen grid topologies, (2) new tasks not in the training set, (3) transfer learning scenarios, or (4) out-of-distribution conditions. Figure 6 shows task embeddings are well-separated after training, but this only demonstrates that the model has learned to distinguish tasks it was trained on—not that it can generalize. The conclusion acknowledges "it remains to explore whether the model performs in a new task or new topology," which should have been addressed in the experiments.

Dataset Scale and Diversity Concerns: Training with only 32k samples per task-grid pair (1.05M total) is modest for a claimed "foundation model." The power-law scaling analysis (Fig. 8) is interesting but the curves haven't saturated, suggesting the model is data-hungry. More critically, all data comes from simulations with synthetic faults and scenarios—the model hasn't been validated on the heterogeneous, noisy, real-world operational data that exists in actual power systems.

Incomplete Ablation Studies: Key design choices lack thorough ablation: (1) Why Conv-1D specifically for the adaptive encoder versus other architectures? (2) How sensitive is performance to the number of GCN layers? (3) What's the contribution of the two-stage training versus end-to-end training? (4) How does performance degrade when certain instructions or graph components are ablated? The only substantive ablation is MoT vs. dense model.

Presentation Issues: The paper has several clarity problems: (1) Figure 2 is overly complex and hard to parse, (2) the relationship between "topology-independent" and "topology-dependent" data (Fig. 9-10) is confusing, (3) critical implementation details are relegated to appendix, (4) the evaluation metrics section (A.6) with its complex reward functions should be in the main paper to understand what "optimality gap" means, (5) inconsistent terminology (e.g., "graph-language answering" vs. "graph-language foundation model").

**Questions:**

See weakness

---

> ### Author Response · Authors · 2025-11-21
>
> We thank the reviewer for their thoughtful comments and suggestions. The following are our responses.
>
> **Overclaim on Foundation Model**. First, we want to politely point out a misunderstanding. Our model is not trained separately on each task-topology pair, but rather on all task-topology pairs together. As OpenGLA has demonstrated adaptability on multiple tasks and topologies, we believe it could serve as a foundation model architecture for power systems if training scales. Considering the current experiment scale, we will consider toning down the foundation model claim in the final revision.
>
> **Unseen topologies**. Please refer to the general response for supplementary experiment details.
>
> **Evaluation on synthetic data**. Topology information of real-world power grids is typically confidential. Although there is open-source real PMU data, it doesn’t come with the topology information. Direct testing in a real power grid is not possible. Therefore, using real loads and synthetic topologies is common practice in power system research. PyPower, PowerWorld, and PSSE are also widely used in both research and actual power system operation.
>
> **Insufficient literature**. We revise the Graph Neural Networks section in the preliminaries to incorporate the up-to-date GNN works.
>
> **Insufficient ablations**.  For Conv1D, it is basically the same in functions as Linear, but more computationally efficient and easier to train when dealing with long sequence input. We add two more ablations: 1) single-stage training vs. two-stage training, 2) changing the number of GNN layers in the graph encoder. Single-stage training would destroy the language capabilities of the language model and cause a catastrophic performance degradation on the fault detection task (language output).
>
> | Single/Two-phase | OPF | Fault_type | Fault_loc | State Estimation | LMP_pred |
> |--------------------------|-----|-------------|------------|-------------------|-----------|
> | IEEE14  | 0.230/0.014 | 0.0/0.963 | 0.130/0.945 | 105.489/1.101 | 2.784/0.012 |
> | IEEE39  | 2.561/0.082 | 0.0/0.844 | 0.128/0.772 | 107.395/1.323 | 1.569/0.013 |
> | IEEE57  | 0.416/0.011 | 0.0/0.867 | 0.131/0.813 | 104.437/0.875 | 2.752/0.315 |
> | IEEE118 | 0.294/0.069 | 0.0/0.855 | 0.184/0.654 | 76.878/3.921 | 11.068/1.142 |
> | IEEE300 | 1.102/0.043 | 0.0/0.834 | 0.110/0.612 | 100.907/1.186 | 3.950/0.165 |
>
> **Presentation issues**. 1) We will improve figure quality in the final PDF revision. 2) ‘Topology-independent’ means the data is irrelevant to power system physics, like load consumption and renewable generation capacity. On the other hand, ‘Topology-dependent’ represents data relevant to power system physics and can only be acquired through physical simulations based on ‘topology-independent’ data. 3) and 4) We don’t want to mess up the methodology section, so we put the data generation part in the appendix. 5) We will correct all terminology in the final revision.
>
> We hope our responses could address your concern.

---

> > ### Comment · Reviewer_JsEo · 2025-11-26
> >
> > Thank you for your response. However I don't think all my questions are answered properly and my concerns remain valid. Therefore I will remain the original scores.

---

> ### Author Response · Authors · 2025-11-29
>
> We would greatly appreciate it if the reviewer could point out what concerns still remain and what responses are not supportive. We summarized the recent revisions in response to your critical concerns as follows.
>
> **Overclaim on the foundation model**. Our model was trained on a dataset spanning multiple tasks and topologies, thus possessing multi-task and cross-topology capabilities. Given our training scale, we prefer to limit our claim to cross-topology and multi-task capabilities, and we have corrected the relevant statements.
> The initial settings were limited to the tasks and topologies in the training set, mainly because zero-shot/few-shot settings might not be realistic given the risk appetite of the power industry, and reducing case-by-case development was a primary request. Based on reviewers' suggestions, we conducted supplementary experiments with N-1 (Table 1) and unseen topologies (Table 2), and acknowledged in the discussion that it could be further improved.
>
> **Usage of synthetic data**. As mentioned before, using synthetic grids and real profiles is a common practice in electric power research considering the confidentiality of the real infrastructure. Publicly disclosed PMU/SCADA data doesn’t contain topology information.
>
> **Insufficient related literature**. As mentioned before, we rewrite the GNN preliminaries to include more up-to-date algorithms. We chose GCN as the main component for graph learning because the power grid does not have overly complex heterogeneous properties, and the number of nodes is much smaller compared to social networks and paper citation networks. We focused more on multi-task design, embedding alignment, and training methods.
>
> **Insufficient baselines**. We added GraphGPT[1] as an advanced baseline for fault detection, since it is capable of the node classification task, however, cannot extend to float regression tasks like OPF, state estimation, and LMP prediction. GraphGPT’s workflow on node classification is similar to the OpenGLA w/o MoT setting, which also aligns the graph embedding with language hidden states then generates language outputs. We also employ a graph transformer model, GPS[2], as the baseline for OPF,  state estimation and LMP prediction. The result could be found in Fig. 4 in the latest PDF revision.
>
> **Scalability issue on Texas2000 fault localization**. Texas2000 has over 22k possible fault type and location combinations. And the number of training samples we used in Texas2000 keeps 32k, the same as the other systems have, which means there are only a few samples for each type-location pair and naturally leads to unsatisfactory performance. However, this could be resolved by simply expanding the dataset, which however, requires strenuous effort to collect new data and is probably not doable within the rebuttal phase. The purpose of our TX2000 experiments was to verify the scalability of the architecture and its performance on simpler classification (fault categories) and continuous regression tasks. The challenging localization tasks require significantly more data.
>
> **Limited architecture innovation**. Our innovation lies in the effective design of graph-language models, regression decoders, and training strategies to achieve multi-tasking, cross-topology ability , laying the architecture basis for power system foundation models.
>
> **Insufficient ablations**. We have added ablations on the number of GCN layers (Fig.12) and single-stage training vs. two-stage training (Table. 4). In summary, insufficient or excessive GCN layers both lead to performance degradation, and single-stage training leads to significant performance drop.
>
> [1] Tang, J., Yang, Y., Wei, W., Shi, L., Su, L., Cheng, S., ... & Huang, C. (2024, July). Graphgpt: Graph instruction tuning for large language models. In Proceedings of the 47th International ACM SIGIR Conference on Research and Development in Information Retrieval (pp. 491-500).
>
> [2] Rampášek, L., Galkin, M., Dwivedi, V. P., Luu, A. T., Wolf, G., & Beaini, D. (2022). Recipe for a general, powerful, scalable graph transformer. Advances in Neural Information Processing Systems, 35, 14501-14515.

---

### Official Review · Reviewer_NQSx · 2025-11-07

**Soundness:** 3
**Presentation:** 3
**Contribution:** 4
**Rating:** 8
**Confidence:** 4

**Summary:**

This work proposes a multi-modal graph–language foundation model for power systems. It integrates a graph encoder, a mixture-of-transformers backbone, and a transformer-based detokenizer to generate both continuous floating-point outputs and language responses. This framework is highly relevant for power-system applications, which are inherently graph-structured and involve diverse downstream tasks. Overall, I consider this work an important and timely contribution to the community.

**Strengths:**

1. The idea of designing a graph–language foundation model is very suitable for power-system tasks, which have varying graph topologies and require multi-task capability across heterogeneous downstream tasks.

2. The model is well designed to accommodate both language-response tasks and exact-value prediction tasks, reflecting a good understanding of the domain requirements.

3. The experimental study and ablation analyses are comprehensive and demonstrate solid empirical validation.

**Weaknesses:**

1. The training data include only six fixed topologies (IEEE 14/39/57/118/300 and Texas2000). Thus, the claimed “topology adaptivity” refers to performance consistency across these seen grids, not to unseen networks. Incorporating N-1 topology perturbations, as explored in CANOS, would significantly strengthen the evidence for robustness to topological changes.

2. The detokenizer converts hidden states into continuous values through autoregressive generation, which may slow down inference. Moreover, Figure 11 reports inference time without ensuring a fixed optimality gap threshold across methods, which makes the comparison less fair. It would be more informative to report the time required to reach the same gap.

3. Since the model learns OPF solutions via supervised regression, it does not guarantee physical feasibility. Discussing potential post-processing or feasibility-repair mechanisms would clarify the model’s practical applicability.

**Questions:**

1. Can you evaluate the inference time at a fixed optimality gap?
2. Can you extend the framework with N-1 contingency or unseen-topology training data to verify robustness to topological perturbations?
3. How does the model handle feasibility violations in predicted OPF or LMP results?
4. Would a parallel decoding or any other methods improve inference time?

---

> ### Author Response · Authors · 2025-11-21
>
> We sincerely appreciate Reviewer NQSx’s insightful and encouraging feedback. The responses to the concerns are as follows:
>
> **Topology generalization**. It seems to be a common concern of all reviewers. Hence, we have an official comment for all reviewers to avoid duplicate responses. Please refer to the official comment for more details.
>
> **Inference time under fixed optimality gap**. Given the inherent performance gaps between different algorithms, it's unlikely that increasing inference time will unify the optimality gap. However, test-time scaling is an interesting idea, which we are willing to discuss in future work.
>
> **Physical feasibility**. For the OPF, we used a simple post-processing to map the normalized output to a reasonable action space, such as maximum/minimum power. Beyond that, we did not perform any remedial constraint processing, such as power balance, reactive power, or voltage constraints.
>
> **Parallel decoding**. This would be a critical issue, especially as power systems scale up to thousands or even tens of thousands of nodes. Serialization prediction can still cause memory overflow and inference speed problems in the model; we will address parallel decoding in the next paper.
>
> We hope our responses could address your concerns.

---

> > ### Comment · Reviewer_NQSx · 2025-11-27
> >
> > Thank you for the clarifications. The authors have addressed my concerns, and I will maintain my positive score.

---

### Author Response · Authors · 2025-11-21
**General response for N-1 and unseen topologies & pdf revision summary**

To all reviewers,
It seems that evaluating generalization capability on topology changes (N-1) and unseen topologies are your common concerns. To avoid duplicate responses, we provide our answers here.

**N-1** is primarily used to evaluate the robustness of the OPF solution to component failures like line outages. In supplementary experiments, we used the output of OpenGLA as the OPF solution and enumerated line failures in subsequent AC power flow simulations. We conducted experiments on the IEEE 14, 39, 57, 118, 300 node systems in the training set, and the results are shown below.

| N-1 OPF pass rate | OpenGLA  | DeepOPF | CANOS |
|--------------|---------------|----------------|------|
| IEEE14  | 97.1% | 91.2% | 95.1% |
| IEEE39  | 85.5% | 80.7% | 83.6% |
| IEEE57  | 98.7% | 89.5% | 92.4% |
| IEEE118 | 89.2% | 72.3% | 85.1% |
| IEEE300 | 92.6% | 76.4% | 87.8% |

As demonstrated, OpenGLA still maintains a pass rate of over 90% on average, a result that is essentially similar to single-scenario solver methods and outperforms baseline algorithms.

**Unseen topologies**, *PGLib89* and *PGLib500* [1] were introduced to evaluate the topology generalization of OpenGLA. We introduced three settings for these three new topologies: zero-shot, transfer learning and full training. The transfer learning setting uses 1k training samples per new topology-task pair. Full training uses the full-size 32k samples for each new task-topology pairs. We found that zero-shot did not lead to catastrophic performance degradation. The performance drop in the fault localization tasks is because that the model outputs are absolute bus indexes, so the model need some examples to align the bus index space. The performance could be significantly improved through a few samples in transfer learning.
| zero-shot/transfer/full   | OPF        | Fault_type   | Fault_loc | State_Est          | LMP_Pred         |
|-----------|-----------|--------------|-----------|--------------|-------------|
| PGLib89   | 0.241/0.159/0.138 | 0.330/0.182/0.159  | 1.0/0.322/0.217    | 12.871/7.336/4.387   | 4.761/1.574/1.103   |
| PGLib500  | 0.832/0.316/0.274 | 0.389/0.217/0.201  | 1.0/0.524/0.278    | 21.534/11.211/5.138  | 8.203/2.135/1.987   |


**PDF revisions:**

We rewrite the GNN section in preliminaries to include up-to-date GNN works.

We include N-1 and unseen topology evaluation results, as demonstrated in Table 1 and Table 2.

We add an ablation about single-stage training vs. two-stage training, as demonstrated in Table 4.

We add an ablation study on the number of GCN layers, as demonstrated in Fig. 12.

We add a minimal qualitative visualization of the fault localization process, as demonstrated in Fig. 13.

We add new baselines, GraphGPT[2] for fault detection, and graph transformer GPS[3] for OPF, state estimation and LMP prediction, as demonstrated in Fig. 4.

[1] Babaeinejadsarookolaee, Sogol, et al. "The power grid library for benchmarking ac optimal power flow algorithms." arXiv preprint arXiv:1908.02788 (2019).

[2] Tang, Jiabin, et al. "Graphgpt: Graph instruction tuning for large language models." Proceedings of the 47th International ACM SIGIR Conference on Research and Development in Information Retrieval. 2024.

[3] Rampášek, Ladislav, et al. "Recipe for a general, powerful, scalable graph transformer." Advances in Neural Information Processing Systems 35 (2022): 14501-14515.

---

### Author Response · Authors · 2025-12-04
**Final comments to AC**

We thank the AC and reviewers for their careful evaluations and for recognizing the importance of a graph-language foundation model tailored to power grid applications. We appreciate that the reviewers found: (i) the motivation for a topology- and task-adaptive graph-language model for the power sector to be clear, (ii) the dual design for language responses and exact floating-point predictions to be well aligned with power system operation requirements, (iii) the experiments to be comprehensive and empirically strong, and (iv) the model performance to be scalable in both parameters and data.

We briefly summarize how the revision and rebuttals address the key concerns raised by reviewers.

**Scope of “topology adaptivity” and robustness to perturbed/unseen grids** [Reviewer NQSx, JsEo, KGmd]

All reviewers correctly pointed out that our training set initially included six fixed topologies (IEEE14/39/57/118/300 and Texas2000), and that our original wording around “topology adaptivity” was interpreted as generalization to perturbed grids like N-1, and fully unseen grids. In the revision, we have:

*Added supplementary experiments on N-1 contingency scenarios and unseen topologies [Sec. 4.2].*

OpenGLA architecture can naturally incorporate such topology changes. In the OPF N-1 contingency demonstrated in Table 1, OpenGLA maintains stable performance under moderate perturbations and outperforms baselines. However, the more challenging N-k problem remains to be explored in future work.

We also evaluated OpenGLA on two unseen grids, PGLib89 and PGLib500[1]. The results show that unseen topologies don't lead to catastrophic performance degradation on OpenGLA’s zero-shot test. Furthermore, after only 1k samples transfer learning, OpenGLA significantly narrows the performance gap relative to full training on the new topologies. We note that future work is needed to explore if zero-shot performance could be further improved through scaling training.

*Clarified the current scope of topology adaptivity.*

The model is originally trained and evaluated across multiple but fixed grids, and now we explicitly distinguish this from zero-shot generalization to unseen topologies. We also included supplementary results and discussions on perturbed and unseen grids as mentioned above.


**Insufficient literature and baselines** [Reviewer JsEo]

Reviewer JsEo raised two related issues: (i) The related works didn’t include up-to-date GNN-related advances, (ii) The baselines didn’t include advanced GNN variants like graph transformer and graph-language models. In response, we have:

*Rewrote the GNN subsection in the preliminaries.* [See Sec. 2.1]

We reviewed the GNN development and included advanced GNN variants, such as GPS[2] for the graph transformer and GraphGPT[3] for the graph-language model.

*Included advanced baselines*. [See Sec. 4.1 and Sec. 4.2]

We added GPS as baseline for OPF, State estimation and LMP Prediction. We also included GraphGPT as baseline for fault detection.


**Insufficient ablations** [Reviewer JsEo, KGmd]

Reviewer JsEo and KGmd proposed two important additions to ablation studies: (i) What is the performance difference between single-stage training and two-stage training? (ii) How will the performance vary as the number of GCN layers changes?

In response, we conducted single-stage training and found a significant performance degradation compared to two-stage training, as demonstrated in Table 4. In addition, we also concluded experiments by setting the number of GCN layers to 1,3,5,7. We found that either shallow or excessive GCN layers would lead to insufficient performance, as demonstrated in Fig.12.

We thank the reviewer for helping improve the comprehensiveness of our experiments.

**Lack of qualitative analysis** [Reviewer KGmd]

Reviewer KGmd suggested to include qualitative analysis, such as visualization showing why the model makes predictions. In response, we added a qualitative visualization on how OpenGLA processes for fault localization on IEEE14 grid, from both graph embedding similarity and attention score perspectives, as shown in Fig.13. The results show that OpenGLA learns a graph embedding that effectively capture nodal abnormal features (high self-correlation and low cross similarity). In the transformer block, the visualized last-layer attention visualization demonstrates that OpenGLA correctly assigns the highest attention to the fault bus.


[1] Babaeinejadsarookolaee, Sogol, et al. "The power grid library for benchmarking AC optimal power flow algorithms." arXiv preprint arXiv:1908.02788 (2019).

[2] Rampášek, Ladislav, et al. "Recipe for a general, powerful, scalable graph transformer." Advances in Neural Information Processing Systems 35 (2022): 14501-14515.

[3] Tang, Jiabin, et al. "GraphGPT: Graph instruction tuning for large language models." Proceedings of the 47th International ACM SIGIR Conference on Research and Development in Information Retrieval. 2024.

---

### Meta-Review · Area_Chair_a6q2 · 2026-01-07

**Summary:**

The paper proposes OpenGLA, a graph language model which aligns topology-adaptive graph encoders with LLMs for power-system tasks. Reviewers appreciated the domain motivation and architecture design and agreed that the needs of the domain well support the idea of foundation modeling across diverse graph topologies, and also appreciate empirical experiments grounded in the domain understanding (NQSx, KGmd). However, reviewers still have major concerns

- claims of being a "foundation model" given reliance on synthetic data, limited scale and lack of convincing zero-shot or cross-topology generalization (JsEo, KGmd)

- weak presentation and literature coverage despite partial revisions (JsEo, AC)

Despite author revisions, the manuscript still has presentation and framing issues which limit the impact of the work. Also, this work could significantly strengthen its contribution by contextualizing and generalizing the insights and architecture to general varying-topology graph foundation modeling / self-supervised learning, rather than focusing only on power systems. This may also make the work more appealing and interesting to a broader class of researchers and uplevel the impact of the work.  The current assessment is that the work is not yet ready for acceptance at ICLR.

**Reviewer Concerns:**

See above.

**Reviewer Scores:**

NQSx: 8->8
JsEo: 2->2
KGmd: 6->6

---

### Decision · Program_Chairs · 2026-01-26

Reject